# Anti-Bacterial Effect of Cannabidiol against the Cariogenic *Streptococcus mutans* Bacterium: An In Vitro Study

**DOI:** 10.3390/ijms232415878

**Published:** 2022-12-14

**Authors:** Tamar Barak, Eden Sharon, Doron Steinberg, Mark Feldman, Ronit Vogt Sionov, Miriam Shalish

**Affiliations:** 1Biofilm Research Laboratory, The Institute of Biomedical and Oral Research (IBOR), Faculty of Dental Medicine, The Hebrew University of Jerusalem, Jerusalem 9112102, Israel; 2Department of Orthodontics, Hadassah Medical Center, Faculty of Dental Medicine, The Hebrew University of Jerusalem, Jerusalem 9112102, Israel

**Keywords:** anti-bacterial, anti-biofilm, cannabidiol, dental caries, *Streptococcus mutans*

## Abstract

Dental caries is caused by biofilm-forming acidogenic bacteria, especially *Streptococcus mutans*, and is still one of the most prevalent human bacterial diseases. The potential use of cannabidiol (CBD) in anti-bacterial therapies has recently emerged. Here we have studied the anti-bacterial and anti-biofilm activity of CBD against *S. mutans*. We measured minimum inhibitory concentration (MIC) and minimum biofilm inhibitory concentration (MBIC). The bacterial growth and changes in pH values were measured in a kinetic study. The biofilm biomass was assessed by Crystal Violet staining and 3-(4,5-dimethyl-2-thiazolyl)-2,5-diphenyl-2H-tetrazolium bromide (MTT) metabolic assay. Spinning Disk Confocal Microscopy (SDCM) was used to assess biofilm structure, bacterial viability and extracellular polysaccharide (EPS) production. CBD inhibited *S. mutans* planktonic growth and biofilm formation in a dose-dependent manner, with similar MIC and MBIC values (5 µg/mL). CBD prevented the bacteria-mediated reduction in pH values that correlated with bacterial growth inhibition. SDCM showed a decrease of 50-fold in live bacteria and EPS production. CBD significantly reduced the viability of preformed biofilms at 7.5 µg/mL with an 80 ± 3.1% reduction of metabolic activity. At concentrations above 20 µg/mL, there was almost no bacterial recovery in the CBD-treated preformed biofilms even 48 h after drug withdrawal. Notably, precoating of the culture plate surfaces with CBD prior to incubation with bacteria inhibited biofilm development. Additionally, CBD was found to induce membrane hyperpolarization in *S. mutans*. Thus, CBD affects multiple processes in *S. mutans* including its cariogenic properties. In conclusion, we show that CBD has a strong inhibitory effect against cariogenic bacteria, suggesting that it is a potential drug adjuvant for reducing oral pathogenic bacterial load as well as protecting against dental caries.

## 1. Introduction

Phytocannabinoids comprise a wide range of chemical compounds, including terpenophenols derived from the *Cannabis sativa* L. plant, such as Δ^9^-Tetrahydrocannabinol (Δ^9^-THC), Cannabidiol (CBD), Cannabichromene (CBC) and Cannabigerol (CBG) [1,2,3]. There is a vast variety of cannabinoids that, despite similar chemical structures, exhibit diverse pharmacological actions, which mostly derives from different interactions with the human endocannabinoid and vanilloid systems [4,5].

Cannabidiol (CBD) is a non-psychoactive phytocannabinoid (Figure 1), the second major active component in the *Cannabis sativa* L. plant [3]. CBD has been studied in association with a wide range of ailments, due to its multiple therapeutic effects together with good safety profile and the lack of psychotropic activity [5,6,7,8]. It possesses strong anti-inflammatory activities as well as having anti-epileptic, anti-emetic, anti-psychotic and anti-convulsive effects [1,6,9,10,11]. Due to that, CBD was found beneficial in treatment of various diseases including cancer, inflammatory, neurodegenerative, autoimmune and cardiovascular diseases [1,5,7,12].

The potential use of cannabidiol in anti-bacterial therapies has recently emerged [9,13]. Several in vitro studies suggest that CBD has bacteriostatic as well as bactericidal effect against various bacterial species, in particular Gram-positive bacteria such as *Staphylococcus aureus*, Methicillin-resistant *Staphylococcus aureus* (MRSA) and *Streptococcus faecalis* [9,14,15,16,17]. Recent research in the dental discipline suggests that CBD has a pronounced time-dependent inhibitory effect on biofilm formation as well as disruption of the mature biofilm of *Candida albicans* [18], although much higher doses were required than those needed for Gram-positive bacteria [14,15]. Stahl et al. [16] showed that CBD significantly decreased the bacterial load in dental plaque when compared to conventional dental hygiene products. CBD in mouthwash was found to be as effective as chlorhexidine mouthwash in inhibiting dental plaque growth [19]. Despite this knowledge, CBD has not been fully studied in the dental discipline.

Dental caries is one of the most common chronic bacterial diseases in humans [20]. It is a biofilm-mediated disease, and its onset is determined by a wide range of risk factors, including genetic, dietary, environmental, and behavioral determinants [20]. Caries is characterized by enamel demineralization, a dissolution process caused by organic acids such as lactic acid, which are produced by the cariogenic bacteria upon fermentation of sugars [20,21]. Two major groups of bacteria are known as main caries generators, namely the mutans *Streptococci* group and *Lactobacilli* species [22,23].

The chosen bacterium for this study is *Streptococcus mutans*. *S. mutans* is a Gram-positive, facultative anaerobic coccus, and a common oral bacterium, especially in association with caries lesions [23,24]. It has multiple virulence factors and special characteristics such as acidogenicity (generation of organic acids) and aciduricity (the ability to survive in acidic environment), which are important for the pathogenesis of dental caries [23,24]. Furthermore, it can adhere strongly to both soft and hard abiotic and biotic surfaces by forming biofilm structures. *S. mutans* expresses extracellular glucosyltransferases (GTFs) and fructosyltransferases (FTFs), two enzymes involved in the production of adhesive glucans and fructans from dietary sucrose, resulting in glucan- and fructan-mediated adhesion [25].

The biofilm is formed via an ordered sequence of events, resulting in a structurally and functionally organized, species-rich microbial community enwrapped in an extracellular matrix [26]. Biofilms can be formed on both biotic and abiotic surfaces, including the gingiva, tooth surfaces, dental crowns, orthodontic devices, dentures and prostheses. Typically, biofilms are less sensitive to anti-bacterial and bacteriostatic agents, due to their unique structure that prevents drug penetration and the presence of sessile bacteria with low metabolic activity [26,27]. Therefore, it is much more difficult to eliminate bacteria immobilized in biofilms in comparison to planktonic-growing bacteria [26,27]. In the dental discipline, where several local and systemic diseases emerge due to the presence of oral biofilms [28,29], it is especially important to find new potential anti-biofilm agents. Therefore, the main aim of our study was to study the anti-bacterial and anti-biofilm activity of CBD against the cariogenic *S. mutans* bacterium.

## 2. Results

### 2.1. CBD Inhibits Planktonic Growth and Biofilm Formation of S. mutans

The first aim was to assess the effect of CBD on *S. mutans* planktonic growth and biofilm formation. It is evident from Figure 2 that CBD inhibits the bacterial growth and biofilm formation, and reduces the metabolic activity of biofilm. No significant effect on either growth or biofilm formation was detected up to 2.5 µg/mL of CBD (Figure 2). However, increasing the CBD dose to 5 µg/mL caused a dramatic decrease of bacterial growth, total biomass and the amount of metabolically active cells in biofilms by 90 ± 6% as compared to control (Figure 2). Thus, the MIC and MBIC of CBD was determined to be 5 μg/mL. It is thus likely that most of the anti-bacterial activity of CBD accounts for its anti-biofilm effect. When reseeding the bacterial culture on plain BHI agar plates after being treated with 5 and 10 μg/mL CBD for 24 h, the bacterial growth recovered, suggesting that CBD has a bacteriostatic effect on *S. mutans*.

### 2.2. CBD Treatment Prevents the Drop in pH Caused by S. mutans

Demineralization of teeth, which characterizes dental caries, is caused by acidification of the oral environment. Therefore, a kinetic study of the pH level in the *S. mutans* culture media was performed simultaneously with a kinetic study of the bacterial planktonic growth. The study showed that CBD at low doses of 0.25 µg/mL and 0.5 µg/mL did not affect bacterial growth during the 24 h of incubation (Figure 3A) which was associated with a pH drop down to 5 (Figure 3B). In contrast, at the dose of MIC and above (5 µg/mL and 10 µg/mL), CBD inhibited bacterial growth during the tested incubation time (Figure 3A), and as a result the pH remained neutral (7.2–7.3) (Figure 3B). Notably, after 4 h of incubation with the 1 µg/mL sub-MIC concentration of CBD, the bacteria grew at a rate almost 2-fold higher than untreated control (Figure 3A), while the pH still remained neutral (7.0) (Figure 3B). CBD at 2.5 µg/mL prevented *S. mutans* growth during the first 10 h of incubation, however, after this period bacteria regained growth, reaching almost the same OD as untreated control after the 24 h of incubation (Figure 3A). Intriguingly, the pH dropped only slightly to 6.5 after a 24 h incubation in the presence of 2.5 µg/mL CBD (Figure 3B), still allowing a more or less neutral environment. This finding suggests that CBD might have a direct anti-acidogenic effect on *S. mutans*. This actually goes along with the anti-metabolic activity of CBD as will be discussed below.

### 2.3. CBD Reduces the Metabolic Activity and Prevents the Recovery of Preformed Biofilm of S. mutans

In the oral cavity, cariogenic bacteria are mostly adhered to the teeth in a mature biofilm form, therefore there is a critical need to eradicate mature biofilms. To test the effect of CBD on preformed biofilms, the bacteria were allowed to form a biofilm for 24 h prior to incubation with increasing concentrations of CBD for another 24 h. At the end of incubation, the metabolic activity of the biofilm-embedded bacteria was measured. The data presented in Figure 4 demonstrate that CBD has a strong dose-dependent inhibition on the metabolic activity as assessed by the MTT assay on the preformed biofilms. At 10 µg/mL, CBD reduced the metabolic activity of preformed biofilms by 50% (Figure 4). Increasing the CBD doses to 20 µg/mL and 50 µg/mL dramatically reduced the metabolic activity of preformed biofilm by 75% and 85%, respectively (Figure 4).

Because oral biofilms have a significant role in the pathogenesis of dental caries, which is a chronic disease that requires a long-term treatment, it was important to understand the duration of the effect of CBD on the treated bacterial biofilms. To this end, the bacteria were allowed to form a biofilm for 24 h and then incubated with CBD for another 24 h, as described above. After the 24 h treatment with CBD, the biofilms were washed with PBS and further incubated with fresh BHI medium supplemented with 2% sucrose for 24 h and 48 h to allow for recovery of the dormant biofilm-embedded bacteria. The metabolic activity was assessed after each recovery time using the MTT assay. The data obtained from these experiments demonstrate that mature biofilms, incubated with up to 10 µg/mL of CBD, were able to almost totally recover after a 24 h incubation in the absence of the drug (Figure 4), while 48 h post-treatment, there was a small, but significant, loss of 20% in the metabolic activity as compared to untreated control (Figure 4). A similar loss of metabolic activity was observed with 5 µg/mL of CBD, while the lower doses of 1 µg/mL and 2.5 µg/mL had no inhibitory effect (Figure 4). A 24 h incubation of a preformed biofilm with CBD at concentrations of 20 µg/mL and above caused an irreversible anti-metabolic effect even 48 h after drug removal (Figure 4).

### 2.4. Precoating of Surfaces with CBD Alters Bacterial Growth and Biofilm Formation

Precoating of surfaces with anti-adhesive/anti-biofilm agents is a valuable strategy for prevention of biofilm formation. As shown in Figure 5, precoating of the surfaces with CBD leads to inhibition of *S. mutans* planktonic growth and biofilm formation. Already, when the 96-well was coated with an amount of CBD equivalent to 3.75 µg/mL, both planktonic growth and biofilm metabolic activity were reduced by around 20% as compared to untreated control (Figure 5). The further increase of precoated CBD doses (equivalent to 7.5–30 µg/mL) caused notable inhibition of bacterial growth in the range of 80–90% (Figure 5). Less pronounced, but still significant reduction of metabolically active cells in biofilms (more than 60% as compared to untreated control) was detected at these CBD doses when the wells were precoated (Figure 5)

### 2.5. CBD Decreases the Number of Live Bacteria and the Amount of EPS Produced in S. mutans Biofilms

The number of live and dead bacteria, and the amount of EPS produced in the biofilms formed in the absence or presence of CBD for 24 h were determined by SDCM following live staining with SYTO 9 (green fluorescence), dead staining with PI (red fluorescence) and EPS staining with Alexa Fluor^647^-conjugated dextran 10,000 (presented in blue fluorescence). The 3D images of the reconstructed biofilm layers demonstrate that the treatment of biofilms with CBD at 4 µg/mL almost totally prevented bacterial attachment to the surface with only a small, almost negligible amount of EPS (less than 5% as compared to control samples with a *p <* 0.05; Figure 6). Notably, at the sub-MIC/MBIC of 2 µg/mL CBD, the amount of EPS and live cells in the biofilm did not significantly change, while there was a significant increase in PI-stained dead bacteria of 20% as compared to control (Figure 6). Some of the PI staining might be due to extracellular DNA released from dying bacteria.

### 2.6. CBD Increases the Membrane Potential of S. mutans

The membrane potential of *S. mutans* was measured using the DiOC_2_(3) potentiometric reagent immediately after exposing the bacteria to CBD. The green fluorescence represents the amount of dye taken up by the bacteria, while the red fluorescence reflects the membrane potential. Already at a sub-MIC of 2.5 µg/mL, CBD caused an obvious increase in the red fluorescence intensity (Figure 7B,C), while the green fluorescence intensity was not significantly altered (Figure 7A,C), indicating an immediate membrane hyperpolarization. Similar, membrane hyperpolarization was observed after treating the bacteria with 5 and 10 µg/mL CBD (Figure 7A–C). A dose of 1.25 µg/mL CBD had no significant effect on the membrane potential (Figure 7A–C).

## 3. Discussion

The oral microflora is abundant with a wide variety of microorganisms, that can often lead to dental and periodontal infections, constitute a clinical problem and also a risk for systemic pathologies [29,30]. A different suggested approach to overcoming bacterial infections is the use of alternative and natural antimicrobial compounds, such as active ingredients of *Cannabis sativa*. Several studies have screened different materials derived from *Cannabis* for anti-microbial properties, to discover new anti-infective agents [9,13]. Numeral cannabinoids have been found to have potent anti-microbial activity against Gram-positive pathogens mainly, such as MRSA isolates [9,14,15]. Cannabidiol (CBD), the main non-psychoactive component of the *Cannabis*, was found in recent studies as highly effective against various Gram-positive bacteria and also against highly resistant species such as *Streptococcus pneumoniae* and *Clostridioides difficile* [15]. In addition, it was found that it has a profound activity on the biofilm, with little propensity to cause resistance, and it was found to be efficient in a topical in vivo infectious model [15]. The CBD-related compound cannibigerol (CBG) was found to have anti-bacterial and anti-biofilm activities against *S. mutans* [31,32,33].

In the oral cavity, many bacterial species can be associated with the pathogenesis of dental caries, but *S. mutans* is considered as one of the most cariogenic bacteria [23,24]. Therefore, the alteration of virulence properties of *S. mutans* is a key objective in the prevention of dental caries.

In the current study, the MIC of CBD towards *S. mutans* was found to be at a dose (5 µg/mL) that is subtoxic to normal Vero kidney epithelial cells, which underwent apoptosis at concentrations above 25 μg/mL CBD (our unpublished data). Thus, there is a therapeutic window where CBD is capable of inhibiting the growth of the pathogen without cytotoxicity toward mammalian cells. Therefore, CBD can be considered as a potential therapeutic agent for caries treatment. Kinetic growth studies showed that CBD at a sub-MIC of 2.5 µg/mL delayed the initiation of the bacterial log and early stationary growth phases, while CBD at MIC = 5 µg/mL and above retained their growth inhibition effect without signs of recovery after 24 h. These findings indicate that exposure of bacteria to sub-MIC of CBD induces a stress response that temporarily impedes cell division.

Acidogenicity, the ability to produce acid from a wide range of fermentable sugars, is an important virulence characteristic of *S. mutans* to initiate caries and the progression of tooth decay [34]. The pH measurements that were performed simultaneously with kinetic growth demonstrated that the effect of CBD at MIC and above MIC on *S. mutans* acid production is seemingly due to the bacterial growth hindrance. However, at the sub-MIC of 2.5 µg/mL, CBD arrested growth for up to 10 h of incubation, and consequently, prevented the pH drop. However, after a 24 h incubation with this dose of CBD where *S. mutans* reached an optical density similar to that of untreated control samples, the pH of the bacterial supernatant still remained neutral, suggesting that CBD might also have a direct effect on inhibiting acid production. Even at the lower sub-MIC of 1 µg/mL CBD, there was a temporarily impairment in the *S. mutans* acidogenicity without affecting its growth. As an analogy, the anti-caries agent resveratrol was able to prevent pH drop by *S. mutans* at sub-MIC values, which was attributed to the suppression of the bacterial glycolytic pathway [35]. Further studies are required to determine whether CBD has a similar effect. Moreover, the ultimate pH values after 24 h of incubation at the sub-MIC of 2.5 µg/mL CBD was much higher than the critical pH value which equilibrates the level of de/remineralization of the tooth [36].

Biofilm formation is a major pathogenic pathway of cariogenic bacteria such as *S. mutans* [37]. Metabolic activity within the biofilm is a significant determinant in terms of pathogenicity, since it is an indicator of growth and enhances production of various virulence factors. The present study demonstrates that the biofilms formed in the presence of CBD, in addition to impaired total biomass formation, appear with a dramatic reduction of metabolic activity.

The SDCM images further confirmed the potent inhibitory effect of CBD on biofilm formation. Almost no bacteria were observed at the MIC/MBIC of CBD = 5 µg/mL, indicating that no biofilm was formed under these conditions. Despite the lack of bacteria, EPS was still observed at this concentration, albeit at much lower level, suggesting that the extracellular GTF and FTF enzymes released by the bacteria were still active, resulting in the observed EPS production. Those enzymes can be secreted from the bacteria and continue synthesizing glucans and fructans independently of the bacteria [25], thereby being responsible for the extracellular matrix. Interestingly, at the sub-MIC of 2.5 µg/mL CBD, the number of dead cells in the biofilm was significantly higher as compared to the control. This observation could be explained by temporary growth arrest at log and early stationary phases at this dose of CBD, which could lead to an increase of dead bacteria after the 24 h of incubation.

Taken together the above findings, it seems that the anti-biofilm effect of CBD occurred mostly due to its inhibitory effect on growth, as a reduced number of planktonic cells limits biofilm formation. This indicates that its anti-bacterial effect is so effective that it does not allow any bacteria to revive and to further form a biofilm.

Bacterial biofilms are permanent inhabitants of the oral cavity. Therefore, it was important to evaluate the effect of CBD on a preformed biofilm. We demonstrated that CBD significantly decreased the metabolic activity of preformed biofilms of the tested bacteria. These findings suggest that CBD has the ability to penetrate preformed biofilms, which is a key trait in pharmaceuticals designed for the oral environment. Additionally, other studies have observed that CBD exhibits a very potent disrupting activity on the preformed drug-resistant and drug-sensitive biofilm of *S. aureus* with a minimum biofilm eradication concentration (MBEC) of 1–4 μg/mL [15]. Our previous study demonstrated that the closely related cannabigerol (CBG) was able to alter the metabolic activity in mature *S. mutans* biofilms, which is attributed to its anti-bacterial activity [32].

Mature biofilm-associated infections are frequently difficult to treat by antibiotics due to low sensitivity of the immobilized bacteria in the biofilms to the drugs, and poor penetration of the drug within the biofilm [38]. Already developed and matured biofilms on various surfaces are challenging to eliminate, which lead to recurrent infections [39]. In contrast to traditional antimicrobials acting on dividing bacteria, CBD, once penetrating an existing biofilm, targets sessile bacteria embedded in the biofilm. Furthermore, using broth dilution serial passage resistance induction assay, it was observed that CBD does not have a tendency to induce resistance against multidrug and methicillin resistance *S. aureus* [15]. Thus, we anticipate that such resistance will not develop in *S. mutans* either. Actually, we have observed that the bacteria recovered from a CBD treatment was still sensitive to a new CBD dose.

In addition to the disruptive activity on mature biofilms, CBD exerted a notable prolonged anti-biofilm effect. The experiment testing the ability of biofilms to recover after CBD exposure showed that the reduction in metabolic activity in *S. mutans* biofilms caused by CBD is sustained, as this activity did not recover after a 24 h and even a 48 h incubation in drug-free medium.

It was reported that repeated treatments might be necessary to prevent the regrowth of *S. mutans* biofilm [40]. Importantly, our findings indicate that a single treatment of CBD has a lasting effect, even after its removal from the bacterial medium, which suggests its great potential in treating dental caries, especially since chronic diseases require prolonged treatment.

Furthermore, precoating of the surface with CBD even at low concentrations (equal and above the equivalent 7.5 µg/mL) caused inhibition of bacterial growth and biofilm formation of *S. mutans* by more than 50%. These results indicate a high adsorption capacity of CBD and maintenance of its biologic stability and activity at least for additional 24 h. Similarly, a strong reduction in bacterial number in dental plaques spread on agar plates precoated with CBD was detected by others [16]. Clearly the interactions between CBD and the solid surface needs further evaluation, as each surface presents different properties.

Multiple mode of action studies point to membrane alteration as one of the primary mechanisms of CBD against bacteria [15]. CBD at very low doses (0.1–0.2 μg/mL) caused significant depolarization of the *S. aureus* cytoplasmatic membrane [15,17]. Our study demonstrates that exposure of *S. mutans* to CBD leads to immediate hyperpolarization of the bacterial membrane, indicating a bacteria-specific effect of CBD. Hyperpolarization has been shown to affect bacterial viability and cell division [41,42,43,44]. CBD already at the sub-MIC of 2.5 μg/mL, caused bacterial membrane instability, which correlates with the increase of dead cells in biofilms treated with this concentration. Our previous work showed that the possible mechanism of *S. mutans* biofilm inhibition by tea polyphenol, epigallocatechin gallate is also associated with bacterial membrane hyperpolarization [45].

## 4. Materials and Methods

### 4.1. Materials

Cannabidiol (CBD) isolate (99% purity) produced by NCLABS (Prague, Czech Republic) was used for the study. A stock solution of 10 mg/mL of CBD in 100% EtOH was prepared and stored at −20 °C. Respective dilutions of ethanol were used as controls in all the experiments.

### 4.2. Bacteria Source and Cultivation

*S. mutans* (UA159) strain from a frozen stock (−80 °C) was inoculated in brain heart infusion (BHI; Acumedia, Lansing, Michigan, USA) broth at a ratio of 1:100 and incubated at 37 °C for 20 h in a humidified incubator in the presence of 5% CO_2_ until reaching an OD_600nm_ of 1.2–1.3. The overnight bacterial suspension was diluted in BHI for planktonic studies, or with BHI supplemented with 2% sucrose for biofilm studies [31,32].

### 4.3. Determination of Minimal Inhibitory Concentration (MIC)

The MIC values of CBD against *S. mutans* were determined using the twofold serial microdilution method based on the CLSI protocol [31,46]. The tested compounds were added to a 96-well plate containing 200 μL BHI medium and initial bacterial OD_600nm_ of 0.05. The range of final concentrations of CBD was 0.25 µg/mL–10 µg/mL. Bacterial inoculum in the medium without the tested compound served as a positive control, whereas the tested compounds in medium without bacteria served as negative controls. The 96-well plate was then incubated at 37 °C in 95% air/5% CO_2_ for 24 h and planktonic growth was quantified by measuring the absorbance at OD_595nm_ using the Infinite M200 PRO plate reader (Tecan Group Ltd., Männedorf, Switzerland). The MIC was determined as the lowest concentration of the tested compound showing no turbidity after a 24 h incubation.

### 4.4. Determination of Minimal Biofilm Inhibitory Concentration (MBIC)

The assay was performed similar to MIC evaluation except that the conditions were changed to a biofilm-formation-inducing environment by the addition of 2% sucrose to the BHI medium. After incubation of bacteria with CBD for 24 h, spent media containing free-floating bacteria were removed by decantation and the wells were rinsed twice with phosphate-buffered saline (PBS, pH 7.4). Total biomass was quantified by crystal violet staining [32,47]. Briefly, 0.1% crystal violet solution was added to the biofilms and the plates were incubated for 20 min at room temperature. Thereafter, the wells were washed twice with DDW to remove unbound dye. The dye bound to the biofilms was dissolved in 200 μL of 33% acetic acid by shaking for 10 min and the biofilm mass was quantified by measuring the absorbance at 595 nm using the Tecan Infinite M200 PRO plate reader.

The metabolic activity of the biofilm was determined quantitatively using a standard 3-(4,5-dimethyl-2-thiazolyl)-2,5-diphenyl-2H-tetrazolium bromide (MTT) reduction assay (Sigma, St. Louis, MO, USA) [32,47,48]. Briefly, 50 μL of a 0.5 mg/mL MTT solution in PBS was added to the washed biofilms followed by a 1 h incubation at 37 °C. The amount of intracellular tetrazolium precipitates representing the metabolic activity of the biofilm-embedded bacteria was quantified spectrophotometrically by measuring the absorbance at 570 nm in a Tecan Infinite M200 PRO plate reader. MBIC was determined as the lowest concentration of the tested compound showing 90% biofilm inhibition compared to the untreated control.

### 4.5. pH Measurements of Planktonic Growing Bacteria

Bacterial suspension of OD_600nm_ = 0.1 in BHI medium was treated with different concentrations of CBD (0.25–10 μg/mL) and incubated at 37 °C, 5% CO_2_ for 24 h. At various time points, the pH of the samples was measured using pH indicator paper strips (MColorpHast, Merck KGaA, Darmstadt, Germany) and the optical density at 595 nm was measured in parallel in a Tecan Infinite M200 PRO plate reader [31].

### 4.6. Effect of CBD on Preformed Biofilms

*S. mutans* biofilms were allowed to mature in BHI + 2% glucose for 24 h at 37 °C in the presence of 95% air/5% CO_2_ in a 96-well plate, as described in Section 4.4. The biofilms were washed twice with PBS, and exposed to CBD at various doses (0.25 –10 µg/mL) in BHI, or in BHI alone for controls for 24 h at 37 °C, 5% CO_2_. The amounts of *S. mutans* remaining in the biofilms were determined quantitatively using the MTT metabolic assay as described above in Section 4.4.

### 4.7. Biofilm Recovery after Drug Removal

In this experiment, the bacteria were allowed to form a biofilm and were then incubated with CBD for 24 h as described above in Section 4.4. Following the 24 h exposure to CBD, the biofilms were washed with PBS and incubated with fresh BHI medium supplemented with 2% sucrose to allow recovery of the remaining biofilm-associated bacteria. The metabolic activity was assessed 24 and 48 h after drug removal by the MTT assay as described in Section 4.4.

### 4.8. Effect of CBD Precoating of Culture Plate Surfaces on Biofilm Formation

100 µL of CBD diluted in ethanol at concentrations of 0–30 µg/mL was added to the wells of 96-well plate, following by slow evaporation of the ethanol under aseptic conditions for 24 h. The precoated CBD wells were then inoculated with *S. mutans* in BHI supplemented with 2% sucrose for biofilm formation and incubated for 24 h at 37 °C in 95% air/5% CO_2_. After incubation, the planktonic growth was quantified by measuring the absorbance at 595 nm using the Infinite M200 PRO plate reader. Next, the spent media containing free-floating bacteria were removed by decantation and the wells were rinsed twice with PBS. The metabolic activity of the remaining biofilm was assessed by MTT assay as described in Section 4.4.

### 4.9. Biofilm Analysis by Spinning Disk Confocal Microscopy (SDCM)

SDCM was used to examine the structure of the biofilm after treatment with CBD, to detect the presence of live/dead bacteria and extracellular polysaccharides (EPS) [33]. The biofilms were grown in the absence or presence of various concentrations of CBD in 12-well tissue culture plates for 24 h, washed twice with PBS, and then stained with 3.3 µM SYTO 9 (Molecular Probes, Life Technologies, Carlsbad, CA, USA), 10 µg/mL of propidium iodide (Sigma, St. Louis, MO, USA) and 10 µg/mL of Alexa Fluor^647^-conjugated dextran 10,000 (Invitrogen, Thermo Fisher Scientific, Eugene, OR, USA) for 20 min at room temperature. SYTO 9 stains both live and dead bacteria, propidium iodide dyes dead bacteria and extracellular DNA, while Alexa Fluor^647^-conjugated dextran 10,000 binds to EPS. The stained biofilms were washed with DDW and fixed with 4% paraformaldehyde for 20 min, and stored in 50% glycerol in DDW. Samples were visualized and images captured under a Nikon Spinning Disk Confocal Microscope (Nikon Corporation, Tokyo, Japan) with 2.5 μm spaces, resulting in three-dimensional images of the bacteria and EPS distribution within the biofilm that were constructed using the NIS-Element AR software. Three random fields for each sample were analyzed. The amounts of total EPS production, live and dead cells in each sample, were calculated according to the fluorescence intensity using NIS-Element AR software. The data were calculated as EPS production/live cells/dead cells in each layer of biofilm (2.5 µm spaces). The relative fluorescence intensity of total EPS production/live cells/dead cells in biofilms treated with CBD is presented as a sum of fluorescence intensity in all layers of the biofilm and compared to untreated control [33,45,48].

### 4.10. Membrane Potential Assay

The immediate effect of CBD on the membrane potential of *S. mutans* was analyzed using the BacLight Membrane Potential Kit (Molecular Probes, Life Technologies, Eugene, OR, USA) according to the manufacturer’s instructions. Overnight *S. mutans* cultures were resuspended in PBS at an OD_600nm_ of 0.3 and exposed to various concentrations of CBD, and immediately thereafter, 3,3′-diethyloxacarbocyanine iodide (DiOC_2_(3)) was added to a final concentration of 30 μM. After 30 min the bacteria were analyzed by flow cytometry (LSR-Fortessa flow cytometer, BD Biosciences) using the 488 nm excitation laser and collecting the data using the green (530 nm) and red (610/620 nm) filters [49]. The BD FACSDiva software was used for the collection of data and the FCS Express 7 software for analyzing the data [49].

### 4.11. Statistical Analysis

All of the experiments were performed in triplicate and repeated thrice. The data are presented as the average of the triplicates from a representative experiment out of three independent experiments, and the SEM is shown. The data obtained from the treated and control samples were calculated in the Microsoft excel program and compared using Student’s *t*-test with a significance level of *p <* 0.05.

## 5. Conclusions

We have shown that the mode of action of CBD against *S. mutans* is multifactorial and attributed to: inhibition of bacterial growth and subsequently hindrance of biofilm formation, diminished biofilm metabolic activity and prevention of bacterial recovery within the biofilms following CBD treatment. Some of these effects can be attributed to the membrane hyperpolarization caused by CBD. The combined anti-bacterial and anti-metabolic effects of CBD contribute to the prevention in pH drop with implications for being a potential adjuvant drug in protecting against dental caries.

## Figures and Tables

**Figure 1 ijms-23-15878-f001:**
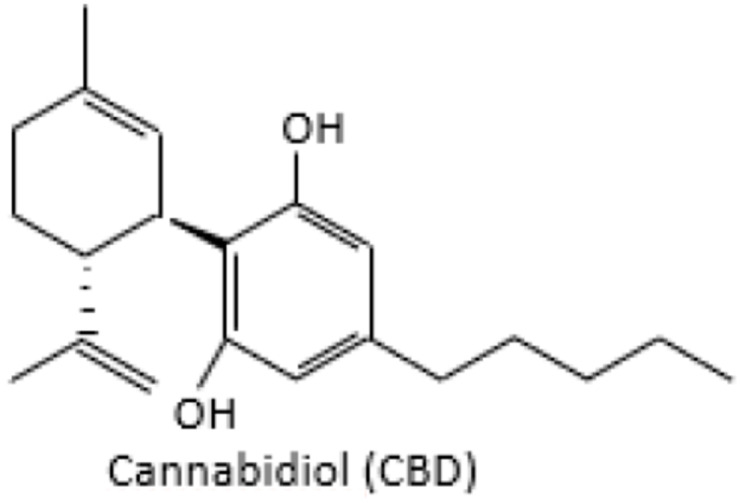
The structure of CBD.

**Figure 2 ijms-23-15878-f002:**
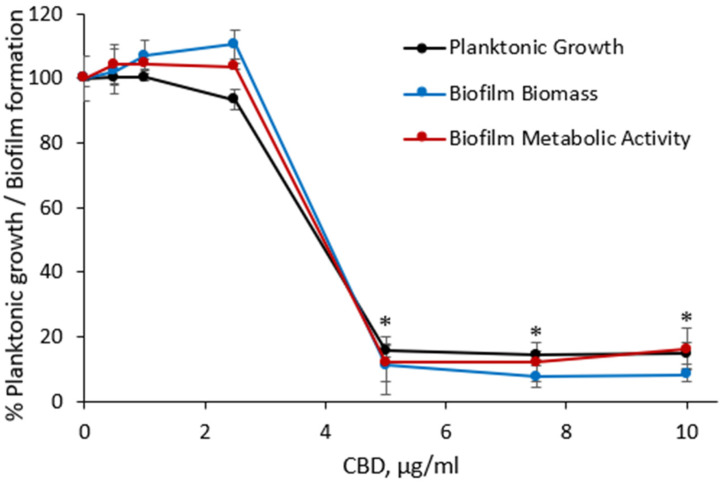
Anti-bacterial and anti-biofilm effect of CBD on *S. mutans*. The bacteria were exposed to increasing concentrations of CBD for 24 h. The bacterial growth was measured at an optical density (OD) of 595 nm. The biofilm biomass was quantified by CV staining and the biofilm metabolic activity determined by the MTT assay. The average and SEM from a representative experiment are presented. * *p* < 0.05 in comparison to the control.

**Figure 3 ijms-23-15878-f003:**
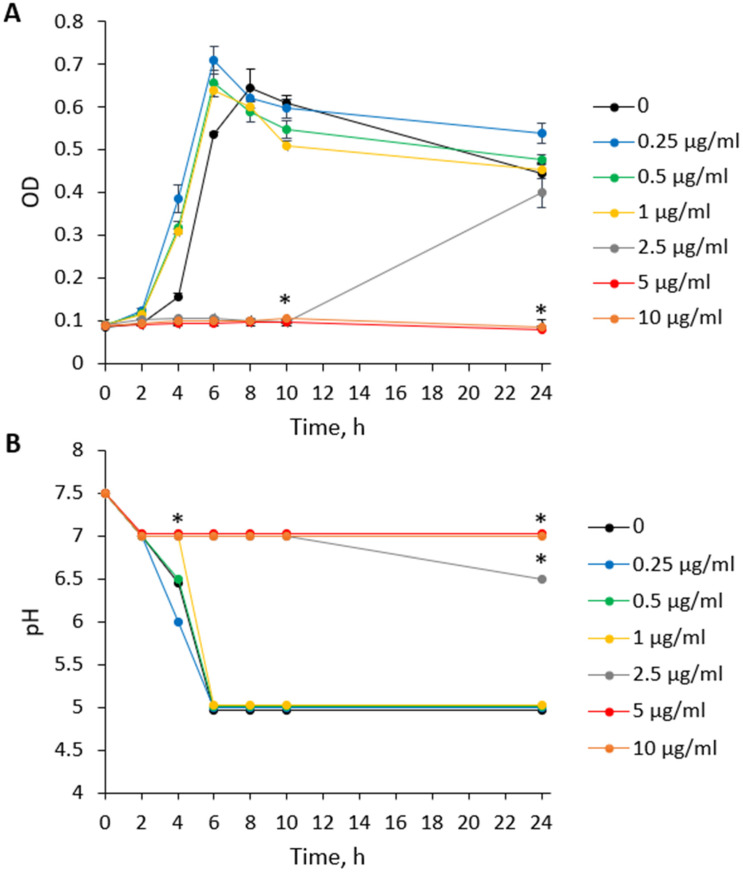
CBD prevents the drop in pH caused by *S. mutans*. (**A**). The growth of *S. mutans* of the same samples used in B. The bacterial growth was measured by an OD of 595 nm at different time points for 24 h. (**B**). The pH of the culture medium of planktonic growing *S. mutans* incubated in BHI with increasing concentrations of CBD. The pH was measured with pH indicator paper strips at different time points for 24 h. The average ± SEM of a representative experiment performed in triplicates is presented. * *p <* 0.05 in comparison to the control.

**Figure 4 ijms-23-15878-f004:**
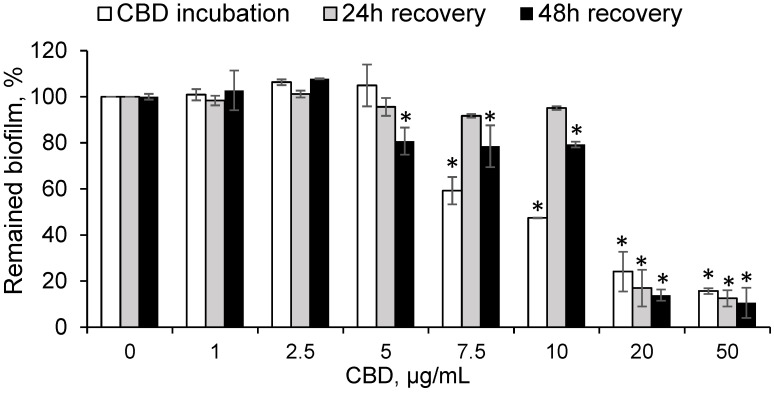
CBD shows prolonged anti-metabolic effect on preformed *S. mutans* biofilms. *S. mutans* were allowed to form a biofilm for 24 h, and then exposed to various concentrations of CBD for another 24 h (white column). Thereafter, CBD was removed, and the biofilms were further incubated in BHI with sucrose for another 24 h (grey column) or 48 h (black column), and the metabolic activity of the biofilm-embedded bacteria was measured by MTT assay at each time point. The average and SEM from a representative experiment are presented. * *p* < 0.05 in comparison to the control of the same time point.

**Figure 5 ijms-23-15878-f005:**
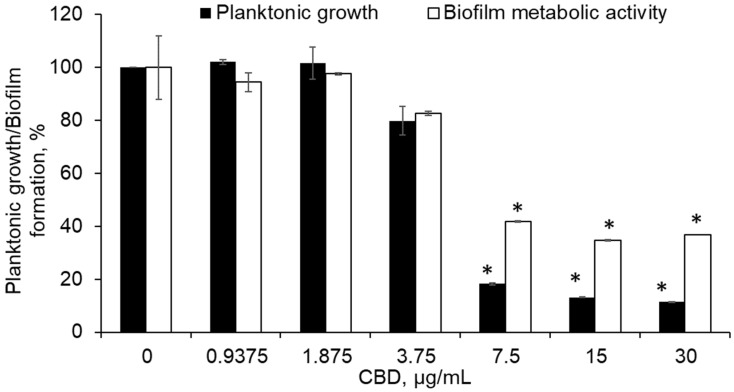
CBD precoating of the culture wells prevented planktonic growth and biofilm formation of *S. mutans*. The bacteria were cultivated on CBD precoated surfaces of a 96-well tissue culture plate under a planktonic or biofilm-forming condition. The bacterial growth was measured at an OD of 595 nm. The biofilm metabolic activity was determined by MTT assay. The average and SEM from three independent experiments are presented. * *p <* 0.05 in comparison to the control of the same time point.

**Figure 6 ijms-23-15878-f006:**
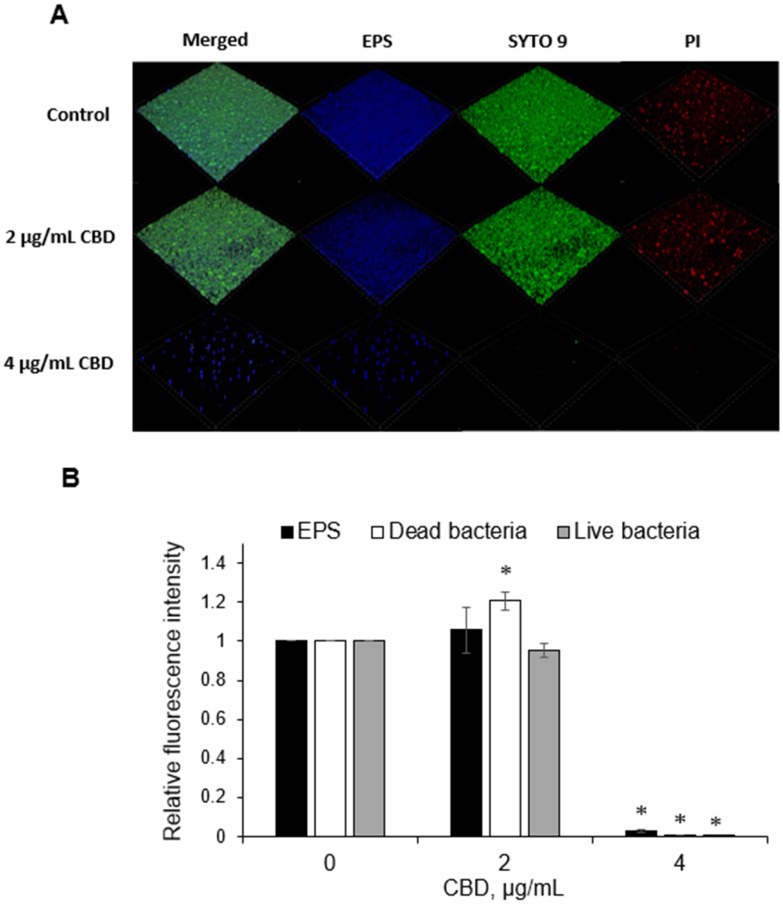
SDCM of live/dead and EPS-stained biofilms. (**A**) SDCM 3D images of merged, live bacteria (green fluorescence), dead bacteria (red fluorescence) and EPS (presented as blue fluorescence). Magnification × 100. (**B**) Quantitative analysis. The amounts of total EPS production, live and dead cells in each sample were calculated according to the fluorescence intensity using the NIS element software. The data are calculated as EPS production/live cells/dead cells in each layer of biofilm (2.5 µm). The relative fluorescence intensity of total EPS production/live cells/dead cells in biofilms treated with CBD is presented as a sum of fluorescence intensity in all layers of the biofilm and compared to untreated control. The average ± SEM of a representative experiment performed in triplicates is presented. * *p <* 0.05 in comparison to the control.

**Figure 7 ijms-23-15878-f007:**
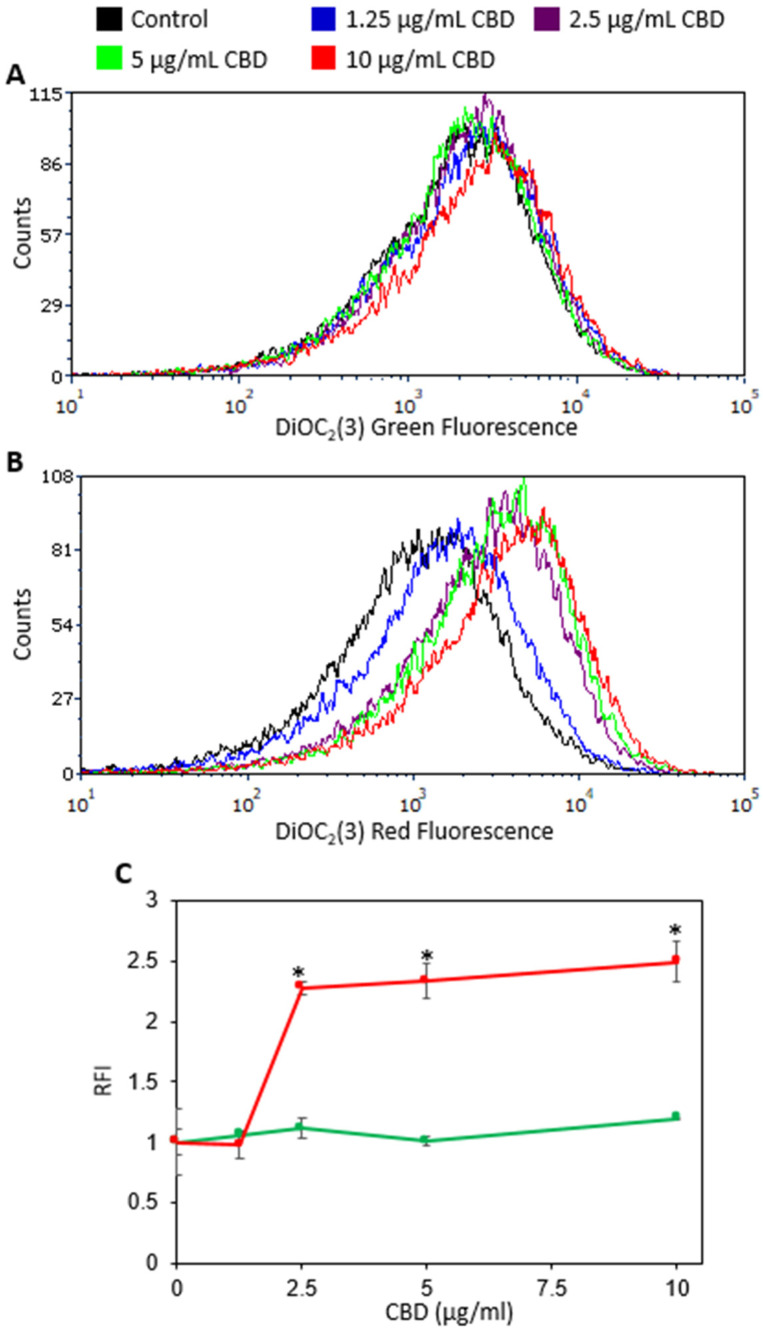
The membrane potential was measured using the DiOC_2_(3) potentiometric dye on flow cytometry immediately after adding CBD. (**A**) The green fluorescence intensity is an indication of the amount of dye taken up by the bacteria. (**B**) Red fluorescence intensity is an indication for the strength of the membrane potential. (**C**) Summaries of the relative fluorescence intensity (RFI) of the red (red lines) and green (green lines) fluorescence for the immediate CBD effect. An increase in the red/green fluorescence ratio is an indication for membrane hyperpolarization. The average ± SEM of a representative experiment performed in triplicates is presented. * *p* < 0.05 in comparison to the control.

## Data Availability

Not applicable.

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
