# Peer review of "Anti-Bacterial Effect of Cannabidiol against the Cariogenic Streptococcus mutans Bacterium: An In Vitro Study"

_ijms, 2022, doi:10.3390/ijms232415878_

Round 1

Reviewer 1 Report

Dear Editor and Authors,

article entitled: Anti-Bacterial Effect of Cannabidiol against The Cariogenic Streptococcus mutans Bacterium is written wery well. I compared this article with previosly study and i must to tell that article is an original and i was not able to find the same study. Therefore i recommend to publish this study with minor correction.  

My opinion: 

Line 102 and 118 – short OD was mentioned firstly, but it was not explained

Line 17 – MTT short, please explain

Line 2 – title of the manuscript does not describe the article accurately, I recommend adding word In vitro, because the reader may think that CBD was applicated directly into the oral cavity, in situ or in vivo, until the material and methods clarified the in vitro study

Line 85 – The last sentence should be start as: Therefore the main of our study was or were… etc

Thank you for opportunity to review 

Author Response

We thank the Reviewer for reading and critically reviewing our manuscript.

article entitled: Anti-Bacterial Effect of Cannabidiol against The Cariogenic Streptococcus mutans Bacterium is written wery well. I compared this article with previosly study and i must to tell that article is an original and i was not able to find the same study. Therefore i recommend to publish this study with minor correction.  

My opinion: 

Line 102 and 118 – short OD was mentioned firstly, but it was not explained

We have now written OD in its full name: Optical density.

Line 17 – MTT short, please explain

We have now written the chemical name of MTT.

Line 2 – title of the manuscript does not describe the article accurately, I recommend adding word In vitro, because the reader may think that CBD was applicated directly into the oral cavity, in situ or in vivo, until the material and methods clarified the in vitro study

We have now added "An in vitro study" to the Title.

Line 85 – The last sentence should be start as: Therefore the main of our study was or were… etc

Thank you for opportunity to review 

We have changed the sentence accordingly to "Therefore, the main aim of our study was to study the anti-bacterial and anti-biofilm activity of CBD against the cariogenic S. mutans bacterium."

Reviewer 2 Report

Interesting article. Raising a very important issue, which is the formation of biofil in the oral cavity. Instead of using drugs, plant substitutes, great. I read it with great attention, however, as a recentent, I have a few comments.

Abstract

The planktonic growth- can better use the growth of microorganisms

Introduction.

It would be good to give the formula of this compound, which is discussed, it always helps to discuss why it has bacteriostatic properties

It would also be good to mention that biofilm also forms on the surface of dentures, which also causes huge problems. Might be an interesting idea for your next article. The surface of the teeth is smoother than the surface of the veneers, where adhesion and formation of biofil occurs very quickly and is very difficult to remove. See: Raszewski, Z.; Nowakowska, D.;Wi˛eckiewicz,W.; Nowakowska-Toporowska, A. The Effect of Chlorhexidine Disinfectant Gels with Anti-Discoloration Systems on Color and Mechanical Properties of PMMA Resin for Dental Applications. Polymers 2021, 13, 1800. https://doi.org/10.3390/polym13111800

After the introduction, there should be material and methods, please correct it.

Results

Line 90

halts the planktonic growth- better to use inhibits the growth of the microorganism

Line 97

seeding the bacteria-culture of bacteria it will be better

Figure 1

Figure 1. Anti-bacteria and anti-biofilm effect of CBD on S. mutans. S. Mutans- reove one S Mutans

What does the abbreviation OD mean, if I'm using it for the first time, please give the full name, please

Figure 2-

What is OD? Planktonic growth- better microorganisms

Line 386

M200 PRO plate reader (Tecan- country?)

Figure 4

CBD precoating of the culture wells prevented planktonic growth and biofilm formation 180 of S. mutans. S. mutans- maybe instead of starting a new sentence with S Mutans, use the strain of bacteria being tested

Line 425

propidium iodide (Sigma- country?)

Line 431

Nikon Spinning Disk Confocal Microscope- country?

Line 442

BacLight Membrane Potential Kit (Molecular Probes, Life Technologies- country?

Good luck with further research

Author Response

We thank the Reviewer for reading and critically reviewing our manuscript.

Interesting article. Raising a very important issue, which is the formation of biofil in the oral cavity. Instead of using drugs, plant substitutes, great. I read it with great attention, however, as a recentent, I have a few comments.

Abstract

The planktonic growth- can better use the growth of microorganisms.

We thank the reviewer for the comment. We, however, prefer to use the well-accepted concept "planktonic growth" in order to distinguish the experiments done in the free-floating state (planktonic growth) and those performed under biofilm formation conditions. In some places, where it was possible, we have changed the "planktonic growth" to "bacterial growth".

Introduction.

It would be good to give the formula of this compound, which is discussed, it always helps to discuss why it has bacteriostatic properties

We have now added the formula of CBD to the Introduction.

It would also be good to mention that biofilm also forms on the surface of dentures, which also causes huge problems. Might be an interesting idea for your next article. The surface of the teeth is smoother than the surface of the veneers, where adhesion and formation of biofil occurs very quickly and is very difficult to remove. See: Raszewski, Z.; Nowakowska, D.;Wi˛eckiewicz,W.; Nowakowska-Toporowska, A. The Effect of Chlorhexidine Disinfectant Gels with Anti-Discoloration Systems on Color and Mechanical Properties of PMMA Resin for Dental Applications. Polymers 2021, 13, 1800. https://doi.org/10.3390/polym13111800

We have accordingly added the following sentence to the Introduction:

"Biofilms can be formed on both biotic and abiotic surfaces, including the gingiva, tooth surfaces, dental crowns, orthodontic devices, dentures and prostheses.”

After the introduction, there should be material and methods, please correct it.

According to the IJMS instructions, the Material and Method Section should appear after the Discussion and before the Conclusion.

Results

Line 90

halts the planktonic growth- better to use inhibits the growth of the microorganism

We have accordingly changed the word to "inhibits".

Line 97

seeding the bacteria-culture of bacteria it will be better

We thank the reviewer for the comment. We have now rephrased the sentence: "When reseeding the bacterial culture on plain BHI agar plates"

Figure 1

Figure 1. Anti-bacteria and anti-biofilm effect of CBD on S. mutans. S. Mutans- reove one S Mutans

We have changed the wordings to "The bacteria".

What does the abbreviation OD mean, if I'm using it for the first time, please give the full name, please

We have now written OD in its full name: "Optical density".

Figure 2-

What is OD? Planktonic growth- better microorganisms

We have now defined the OD in Figure 1. And corrected to "The growth of"

Line 386

M200 PRO plate reader (Tecan- country?)

The following is written at the first time Tecan plate reader is mentioned: "(Tecan Group Ltd. Männedorf, Switzerland)." Thereafter we changed the wordings to: "Tecan Infinite M200 PRO plate reader".

Figure 4

CBD precoating of the culture wells prevented planktonic growth and biofilm formation of S. mutans. S. mutans- maybe instead of starting a new sentence with S Mutans, use the strain of bacteria being tested

We have rephrased the sentence to: " The bacteria were cultivated on CBD precoated surfaces of a 96-well tissue culture plate under planktonic or biofilm forming condition."

Line 425

propidium iodide (Sigma- country?)

The location of Sigma has been added.

Line 431

Nikon Spinning Disk Confocal Microscope- country?

The location of Nikon Corporation has been added.

Line 442

BacLight Membrane Potential Kit (Molecular Probes, Life Technologies- country?

The location of Molecular Probes has been added.

Reviewer 3 Report

Nice work with all the method well described.

Please just check:

line 77 removed dash after structurally

line 210, fig 6, line 445: DiOC2(3) in formula DiOC2(3).

line 217 : 1,25 in place of 1 microg/mL

Line 353: a bracket is missing (UA159).

Author Response

We thank the Reviewer for reading and critically reviewing our manuscript.

Nice work with all the method well described.

Please just check:

line 77 removed dash after structurally

We have now removed the dash.

line 210, fig 6, line 445: DiOC2(3) in formula DiOC2(3).

We have now put the 2 in subscript both in the figure and in the text.

line 217 : 1,25 in place of 1 microg/mL

Thanks for noting this. We have now corrected it.

Line 353: a bracket is missing (UA159).

We have now added the bracket.
